# Evaluation of the Relation between Lean Manufacturing, Industry 4.0, and Sustainability

**Leonilde Varela [1],\*, Adriana Araújo [1], Paulo Ávila [2], Hélio Castro [2] and Goran Putnik [1]**

[1]  Department of Production and Systems, School of Engineering, University of Minho; 4804-533 Guimarães, Portugal; dricafaraujo@hotmail.com (A.A.); putnikgd@dps.uminho.pt (G.P.)

[2]  School of Engineering and CIDEM Research Center, Polytechnic of Porto, 4249-015 Porto, Portugal; psa@isep.ipp (P.Á.); hcc@isep.ipp.pt (H.C.)

\*  Correspondence: leonilde@dps.uminho.pt; Tel.: +351-253-510-765

**Abstract:** Nowadays, Lean Manufacturing, Industry 4.0, and Sustainability are important concerns for the companies and in a general way for the society, principally, the influence of the two production philosophies, Lean Manufacturing and Industry 4.0, in the three main pillars of sustainability: economic, environmental, and social. According to the literature review done in this work, these relations are not well known and are dispersed by different sustainability's criteria. To address this gap, this research proposes a structural equation model, with six hypotheses, to quantitatively measure the effects of Lean Manufacturing and Industry 4.0, in Sustainability. To statistically validate such hypotheses, we collected 252 valid questionnaires from industrial companies of Iberian Peninsula (Portugal and Spain). Results show that: (1) it is not conclusive that Lean Manufacturing is correlated with any of the sustainability pillars; and (2) Industry 4.0 shows a strong correlation with the three sustainability pillars. These results can contribute as an important decision support for the industrial companies and its stakeholders, even because not all the results are in line with other opinions and studies.

**Keywords:** Lean Manufacturing; Industry 4.0; sustainability; economic; environmental; and social; structure equations modeling

## 1. Introduction

At present, Lean Manufacturing (LM), Industry 4.0 (I4.0), and Sustainability are important concerns for companies and in a general way for the society. The influence of the two production philosophies, LM and I4.0, in the three main pillars of sustainability—economic, environmental, and social—for industrial companies situated in Iberian Peninsula (Portugal and Spain) is the main objective of this work. More precisely, [The principal reasons and motivations to develop this study are related with: (1) There is no existing study using the structural equations modeling technique for both production philosophies (LM and I4.0) and the pillars of sustainability; (2) because the knowledge of these potential correlations can influence important decisions for the industrial companies and its stakeholders. In fact, the topic of this work concerns all parts of society minimally affected by the outcomes of LM and, more recently, with I4.0. In relation to this, the uncertainty is large, and few concerns are now appearing from different sides of the society, e.g., related with the future of the employment.

Lean manufacturing (LM), or lean production, in time philosophy, Toyota production system, or more often just "Lean", is a philosophy which considers the utilization of resources for any goal with value creation for the end consumer. It targets the elimination of wasteful activities involved in the value system [1,2]. LM is supported by a set of well-known tools to operationalize its goals, either

at a strategic level or at an operational level, and the basis of the philosophy considers the human being as an import issue in all its decisions. Unfortunately, nowadays the companies that introduce LM practices tend to forget this human aspect, and principally focuses on waste reduction, a side that has brought well known results for the production systems.

The Industry 4.0 or I4.0 for short [3–6], is starting to revolutionize communities requiring a significant upgrade not just in terms of technology. With the advent of exponential technology and high speed and big data processing capabilities, high levels of digitalization regarding all kind of processes in companies are also required. These processes have to become supported by appropriate infrastructures, such as: IoT, IIoT, RFID, CFS, and Cloud, [3–10] along with additional fitting hardware and software means for enabling a full vertical and horizontal integration of all companies' functions, from the administrative level down to the shop floor. Additive manufacturing [9] and collaborative robots [11], for instance, are expected to play a crucial role in this direction, but also suitable organizational structures and business models, and along with appropriate production and decision methods and supporting tools are going to be necessary to enable a successful ingress on I4.0. Moreover, according to [6,12] the principles of Industry 4.0 are the horizontal and vertical integration of production systems driven by real-time data interchange and flexible manufacturing to enable customized production. Such data plays a crucial role for enabling different kinds of decision making, for instance regarding the prioritization of production orders, and tasks optimization, along with other needs, such as maintenance related to each one's requirements [13].

The concept of sustainability has received increasing global attention from the public, academic, and business sectors. The World Commission on Environment Development (WCED) defined sustainable development as "development that meets the needs of the present without compromising the ability of future generations to meet their own needs" [14]. Putnik and Ávila in their special issue of governance and sustainability [15] reinforce the importance of the theme and even give the character of ubiquity in the word 'sustainability'. Nidumolu et al., in 2009 [16] explains why sustainability is now the key driver of innovation according with their study of sustainability initiatives of 30 large corporations. Almeida et al., in 2016 [17] says even that it is common to ignore the interdependence of the sustainability pillars for short periods of time, but history has shown that before long, mankind is reminded of it through some types of alarms or crisis.

In spite of everyone knowing the 3 pillars of sustainability (economic, environmental, and social), it is quite difficult to choose the criteria/KPI to characterize and evaluate the degree of sustainability of each organization. Because our work intends to analyze the relation between LM and sustainability, and I4.0 and sustainability, the criteria that characterize the three pillars of sustainability are crucial to develop the current research. Through a literature search related with the theme were found several works covering different kinds of applications areas, such as agriculture, civil engineering, manufacturing, energy production, and mining. However, only few are framed with manufacturing applications, which is the principal area of our study. To refer those more relevant to our work, the study of a set of researchers [18–24] was the basis to define the criteria for the pillars of sustainability. In fact, there is no existing universal model to assess the level of sustainability for a manufacturing system, and the models are different. For that reason and considering the combination/integration of both methodologies to define criteria for sustainability assessment, strongly recommended by Waas et al., in 2014 [25]: the "top-down"/"expert-driven" and "bottom-up"/"stakeholder-driven", because a combination of the different kinds of knowledge, from the citizen to the experts, defined the criteria to assess the sustainability as used in the Table 1 Table 2 Table 3 Table 4 Table 5 Table 6.

To achieve the goal of this work, the rest of this paper is organized as follow. Section 2 identifies the main contributions of scientific and technical works in the relations between LM and Sustainability, and, I4.0 and Sustainability. Section 3 presents the general model used to study the problem and then is made the evaluation of the model and the results discussion in the Sections 4 and 5 respectively. To finalize, some conclusions and future works are made in Section 6.

## 2. Literature Review

The literature review presented in this section aims at serving as a basis for supporting the main purpose of this work, that will consider two main issues: (1) to accurately regard the theme of this work; (2) the knowledge needed to specify the main constructs of our proposed Structural Equation Modeling (SEM), further described in Section 4.

Regarding issue (1), the purpose is to identify if there are similar concerns among researchers' proposals, and if their works overlap our research proposal.

Assuming that issue 1 is accomplished, as it will be further exposed in this paper, the issue (2) is carried out through the identification of the main relations between LM and Sustainability, and I4.0 and Sustainability. In order to proceed with such identification are considered different kind of studies, including different statements of the researchers reviewed regarding this subject. This statements analyzed acted as a basic knowledge for the definition of the constructs of our proposed model. Therefore, we constructed focused tables to resume the main relations for further analysis through the constructs underlying our proposed model.

### 2.1. Lean Manufacturing and Sustainability

This sections discusses the relations between lean manufacturing (LM) and the 3 dimension of sustainability is the scope of this section. Several organizations have successfully achieved better results and higher competitiveness through LM implementation; however, others have not, as they were unable to sustain medium- and long-term results [26]. Companies that have adopted LM to improve their results also want to be seen as socially responsible. Sustainability is considered the new LM frontier [27]. Productivity and cost-saving are necessary for the economic survival of organizations. However, these tasks should be achieved in a sustainable way, by mitigating negative environmental and social impacts and contributing to a sustainable society [28].

Therefore, in the following Tables 1–3, some main recent contributions, arising from a set of researchers in this cross LM—Sustainability area, and underlying main influences considered are summarized.

### 2.1.1. Influence of Lean Manufacturing in Economic Dimension

Examples of initiatives leading to cost savings and process performance are many attending the influence of LM in the economic dimension. However, for the rest of the criteria considered, the references are scarce and even for the turnover influence were not found references.

In Table 1, a resume about some main contributions regarding the relation or influence of LM in the economic dimension of sustainability are presented.

**Table 1.** Influence of Lean Manufacturing in economic dimension.

| Dimension | Influence | References |
| --- | --- | --- |
| Economic | Increase profits | Pampanelli et al. (2014) [28] |
| | Increase turnover | Not identified |
| | Increase market share of the products | Wilson (2010) [29] |
| | Decrease operational costs | Zhu, et al., 2008 [30]; Mollenkopf et al., 2010 [31]; Sezen et al., 2011 [32]; Lozano and Huishingh, 2011 [33]; Azevedo et al., 2012 [34]; Díaz-Reza, et al., 2016 [35]; Gupta, et al., 2018 [36]; |
| | Increase process performance | Shah and Ward, 2007 [37]; Sezen, et al., 2011 [38]; Ng, et al., 2015; Díaz-Reza, et al., 2016 [35]. |

2.1.2. Influence of Lean Manufacturing in Environmental Dimension

According to Jabbour et al. [38–40] support for environmental management tends to be greater when companies adopt LM practices, which would improve their environmental performance. Also Ng et al., in 2015 [41] refer that LM reduce environmental impact and increase environmental benefits. According to Yang et al., in 2011 [42], that explored the relationship between LM practices, environmental management, and business performance, the results of their research propose that lean manufacturing experiences are positively related to environmental management practices. In spite of that some authors refer the good influences of LM in Sustainability, for the influence of production of renewal energy and for the influence in collaboration with partners that follow good environmental practices were not found references.

In Table 2, a resume about some main contributions regarding the relation or influence of LM in the environmental dimension of sustainability are presented.

**Table 2.** Influence of Lean Manufacturing in the environmental dimension.

| Sustainability Dimension | Influence | References |
| --- | --- | --- |
| Environmental | Decrease industrial waste | Souza and Alves, 2017 [26]; Wilson, 2010 [29]; Torielli, et al., 2011 [43]; Vinodh, et al., 2011 [44]; Gupta, et al., 2018 [36]; Azevedo, et al., 2012 [34]; Hajmohammad, et al., 2013 [45]. |
| | Decrease energy consumption of non-renewal energy sources | Ioppolo, et al., 2014 [46]. |
| | Increase the production of renewal energy | Not identified |
| | Increase the practice of circular economy | Nunes and Bennett, 2010 [47]; Zhao and Chen, 2011 [48]; Ming and Xiang, 2011 [49]; Ashish, et al., 2011 [50]; Liao, et al., 2013 [51]. |
| | Increase the collaboration with partners that follow good environmental practices | Not identified |

2.1.3. Influence of Lean Manufacturing in Social Dimension

The influence of LM is one of the major concern of our study and in the beginning of its implementation inside the Toyota production system. Between the seven major gaps identified by Cherrafi et al., in 2016 [52] in the conclusions of their research, two of them are clearly framed with the goal of our study: the need to study the human side in a more comprehensive manner, and the need to develop an integrated metrics and measurement system to measure the relation between lean and sustainability performance. Gupta et al. [36], in their work related to environmental sustainability, refer also to how future studies could bring the social dimension. In spite of some concerns related to the social pillar, most of the influences did not find any reference that reflects with certainly the low importance that has been given to the topic of sustainability.

In Table 3, some main contributions regarding the relation or influence of LM in the social dimension of sustainability are presented.

**Table 3.** Influence of Lean Manufacturing in social dimension.

| Dimension | Influence | References |
|---|---|---|
| Social | Increase the number of employees | Not identified |
| | Increase the salary remuneration | Not identified |
| | Increase the quality of work conditions | Ng, et al., 2015 [41]; Taubitz, 2010 [53]; Lozano and Huishingh, 2011 [33]; Vinodh, et al., 2011 [44]; Ioppolo, et al., 2014 [46]; |
| | Increase the conditions of the surrounding society | Not identified |
| | Decrease working accidents | James, et al., 2013 [54]; |
| | Increase the participation of its employees in decision-making | Taubitz, 2010 [53]; Vinodh, et al., 2011 [44]; Jabbour, et al., 2012 [39]. |
| | Increase the number of employees with some degree of disability. | Not identified |
| | Increase the contract duration of its collaborators | Not identified |

## 2.2. Industry 4.0 and Sustainability

In this upcoming era of the I4.0 as many changes are expected to occur in the everyday life of people and companies [40], one important and big question we now have to face, among several others, is: "Can industry 4.0 revolutionize the environmentally-sustainable manufacturing wave? In order to make a contribution in this direction, in this work we intend to further explore several contributions and opinions arising from different authors and sources to try to analyze some main positive and negative impacts that I4.0 may have in terms of the three main dimensions of the sustainability concept—economic, social, and environmental—in the context of Industrial companies.

Therefore, in the following Tables 4–6, some main recent contributions, arising from a set of researchers in this cross I4.0— the sustainability area and underlying main dimensions considered are summarized.

### 2.2.1. Influence of I4.0 in Economic Dimension of Sustainability

It is expected that I4.0 will drive companies, for instance industrial ones, to more favorable economic situations, though massive savings to be reached by reducing operators or man power, by saving energy, and by doing work effectively and efficiently, and by reducing production time and improving productivity, among other beneficial situations [11,40,55,56].

In Table 4, a resume about some main contributions regarding the relation or influence of I4.0 in the economic dimension of sustainability are presented.

**Table 4.** Influence of Industry 4.0 (I4.0) in economic dimension.

| Dimension | Influence | References |
|---|---|---|
| Economic | Increase: profits, value creation, efficiency, flexibility, and competitiveness | Müller, et al., 2018 [56]; Nagy, et al., 2018 [57]; Laudien, et al., 2017 [58]; Rennung, et al., 2016 [59]; Erol, et al., 2016 [5]; Rehage, et al., 2013 [60]; Rudtsch, et al., 2014 [61]; Brettel, Klein, and Friederichsen, 2016 [62]; Stock and Seliger, 2016 [63]; |
| | Increase turnover, and create new business models | Arnold, et al., 2015 [64]; Brettel, et al., 2014 [62]; Burmeister, et al., 2016 [65]; Hofmann and Rüsch, 2017 [66]; Duarte and Cruz-Machado, 2017 [67]; Bechtsis, et al., 2017 [68]; de Sousa Jabbour, et al., 2018 [40]; Gilchrist, 2016 [7]; Branke, et al., 2016 [66]; Schmidt, et al., 2015 [67]; Schmidt, et al., Branke, et al., 2016 [69]; 2015 [70]; Nagy, et al., 2018 [57]; Glas, et al., 2016 [71]; |

**Table 4.** *Cont.*

| Dimension | Influence | References |
|---|---|---|
| | Improve: market share of the products, supply chains, and its management performance and security | Dubey, et al., 2017 [72]; Branke, et al., 2016 [69]; Hofmann and Rüsch, 2017 [66]; Stock and Seliger, 2016 [63]; Tjahjono, et al., 2017 [73]; Sommer, 2015 [74]; Wang, et al., 2015 [20]; Lee, Kao, and Yang, 2014 [13]; Luthra and Mangla, 2018 [75]; Nagy, et al., 2018 [57]; |
| | Decrease operational costs | Shrouf, et al., 204 [4]; Waibel, et al., 2017 [76]; Yang, 2014 [13]; Schmidt, et al., 2015 [70]; Stock and Seliger, 2016 [63]; |
| | Improve processes performance, increase renewable resources, and improve circular economy | Jabbour, et al., 2017 [40]; Oettmeier and Hofmann, 2017 [77]. |

### 2.2.2. Influence of I4.0 in Environmental Dimension of Sustainability

In terms of environmental impact, I4.0 may have either positive or negative impacts, depending on several different kinds of analysis that may be carried out, across small to big enterprises [40,66,78].

In Table 5, a resume about some main contributions regarding the relation or influence of I4.0 in the environmental dimension of sustainability are presented.

**Table 5.** Influence of I4.0 in environmental dimension.

| Sustainability Dimension | Influence | References |
|---|---|---|
| Environmental | Decrease industrial waste | Shrouf, et al., 2014 [4]; Waibel, et al., 2017 [76]; Yang, 2014 [13]; Oettmeier and Hofmann, 2017 [77]; Stock and Seliger, 2016 [63]; Wang, et al., 2015 [20]; |
| | Decrease energy consumption of non-renewal energy sources | Hofmann and Rusch, 2017 [66]; Fritzsche, et al., 2018 [79]; |
| | Increase production of renewal energy | Lund, and Mathiesen, 2019 [80]; |
| | Increase practice of circular economy | Jabbour, et al. (2017) [40]; Branke, et al., 2016 [66]; |
| | Increase collaboration with partners that follow good environmental practices | Zawadzki and Żywicki, 2016 [78]; Hofmann and Rüsch, 2017 [66]; |
| | Decrease resources consumption, global warming, climate changes, and energy requirements | Tseng, et al., 2018 [81]; Fritzsche, et al., 2018 [79]. |

### 2.2.3. Influence of I4.0 in Social Dimension of Sustainability

People in general and operators in particular seem to become increasingly worried about the upcoming increasing era of I4.0 due to many reasons, mainly in regard of work opportunities [57]. Although, some more optimistic ones are even trying to foresee very beneficial conditions and opportunities to workers and people in general [55,62,69,82,83].

In Table 6, a resume about some main contributions regarding the relation or influence of I4.0 in the social dimension of sustainability are presented.

**Table 6.** Influence of I4.0 in social dimension.

| Dimension | Influence | References |
|---|---|---|
| Social | Increase number of employees | Branke, et al., 2016 [69]; Brettel, Klein, and Friederichsen, 2016 [62]; |
| | Improve working conditions (e.g., for employees with some disability, training courses, salary, among others) | Shamim, et al., 2016 [82]; Hirsch-Kreinsen, 2014 [83]; Kiel, et al., 2017 [55]; |
| | Improve conditions of the surrounding society | Branke, et al., 2016 [66]; Shamim, et al., 2016 [82]; |
| | Decrease working accidents | Brettel, Klein, and Friederichsen, 2016 [62]; |
| | Increase participation of employees in decision-making | Branke, et al., 2016 [69]; Brettel, Klein, and Friederichsen, 2016 [62]. |
| | Increase contract duration of employee and collaboration among stakeholders | Yang, 2014 [13]; Duarte and Cruz-Machado, 2017 [67]; Pfohl, et al., 2017 [84]; Shamim, et al., 2016 [82]. |

## 2.3. Main Remarks from State-of-the-Art Research

We proposed to postpone the analysis regarding the tables presented for this subsection, because to develop our model it is necessary to have a global analysis over the main contributions arising from state-of-the-art research. Therefore, the main remarks that should be pointed out are the following:

- The first remark is that none of the researchers in their works did cover all the considered main influence criteria exposed in Tables 1–6;
- The second remark is that none of the works analyzed treats this subject through SEM;
- The third remark is that some criteria's influence are more considered than others, namely, for few of them were not found any reference.

Furthermore, we can realize that the two main issues presented at the beginning of this section were accomplished through the summarized review presented on the Tables 1–6. In short, this work focuses a different approach than that the literature, and puts forward the main constructs to be used in our proposed model, as described in the next section.

## 3. Research Model

This section presents the global framework of our model and the methodology used in this work.

### 3.1. General Model

In this work, to evaluate the relationship between LM and Sustainability, and, I4.0 [85–87] and sustainability—the problem of our study—were defined two main sets of hypotheses. The first set of hypotheses is related to LM and Sustainability (H1, H2, and H3), and the second one is related to I4.0 and sustainability (H4, H5, and H6), which are related to the economical (EcS), environmental (EnS), and social (SoS) pillars, as shown below.

**Hypothesis 1 (H1).** *The industrial companies' perception on Economic Sustainability is positively related to Lean Manufacturing.*

**Hypothesis 2 (H2).** *The industrial companies' perception on Environmental Sustainability is positively related to Lean Manufacturing.*

**Hypothesis 3 (H3).** *The industrial companies' perception on Social Sustainability is positively related to Lean Manufacturing.*

**Hypothesis 4 (H4).** *The industrial companies' perception on Economic Sustainability is positively related to Industry 4.0.*

**Hypothesis 5 (H5).** *The industrial companies' perception on Environmental Sustainability is positively related to Industry 4.0.*

**Hypothesis 6 (H6).** *The industrial companies' perception on Social Sustainability is positively related to Industry 4.0.*

By the present model, it is defined that LM and I4.0 are independent variables (exogenous constructs), therefore, these variables do not have an arrow pointing to it from another construct, and EcS, EnS, and SoS are variables dependents (endogenous variables), therefore, the mentioned constructs have a least one arrow pointing to it from another construct. As can be seen, the five constructs (two exogenous and three endogenous) and the six working hypotheses are graphically represented in Figure 1 that represents our initial general model.

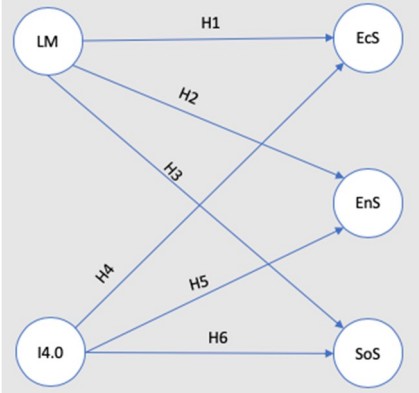

**Figure 1.** Initial general model.

In order the need to represent theoretical models that allow the identification of causal and/or hypothetical relationships between variables, and to validate the theoretical premise that the present work intends to prove, namely, the relation between the manufacturing concepts Lean Manufacturing and Industry 4.0 with Sustainability (Economic, Environment, and Social), we established the use of the modeling technique called Structural Equation Modeling (SEM) [88,89]. This methodology is identified as multivariate analysis, usually expressed in linear models that include measurement errors associated with the established variables in the model [89–91].

In this work, the analysis is divided into two parts the Confirmatory Factor Analysis (AFC) for the measurement model and the structural model analysis. So, based in the SEM method, it is established by two models [86,88]: Measurement Model and Structural Model. The Measurement Model (MM) establishes the relationships between the constructs and its manifested variables, in which the construct is formed by manifested variables through the accomplishment of the Confirmatory Factor Analysis (AFC) that calculates and specifies how the constructs are measured from the manifested variables. In the Structural Model (SM) the relationship between exogenous and endogenous constructs is defined. In this model is established the influence (direct or indirect) that the exogenous constructs apply on the endogenous constructs. In the development of the present work, it was used the software IBM SPSS Amos, version 24, from IBM Corp., 2016 [89] to employ the SEM method.

*3.2. Survey and Data Collection*

In this study a set of variables were established (see Table 7) based on the literature review carried out and summarized in Section 2, and our main findings for the considered problem.

**Table 7.** Constructs and manifested variables.

| Constructs | Manifested Variables |
| --- | --- |
| Exogenous | |
| Lean Manufacturing (ξ1) | Pull production (X1) |
| | Product defects (X2) |
| | Failures (X3) |
| Industry 4.0 (ξ2) | Big data (X4) |
| | Autonomous robots (X5) |
| | Digitalization (X6) |
| Endogenous | |
| Economic Sustainability (η1) | Profits (Y1) |
| | Turnover (Y2) |
| | Market share (Y3) |
| Environmental Sustainability (η2) | Energy consumption (Y4) |
| | Circular economy (Y5) |
| | Environmental practices with partners (Y6) |
| Social Sustainability (η3) | Salary remuneration (Y7) |
| | Work conditions (Y8) |
| | Surrounding society (Y9) |

Next, a survey based on these premises was developed, validated by three experts, and put available to a widened industrial community of the Iberian Peninsula. The survey was applied only in industrial companies located in Portugal and Spain, and it was collected 252 validated answers in a total of 306 answers obtained. During the survey, the respondent answered in a 5-points Likert scale in which 1 means the respondent completely disagree with the statement (lowest value) and 5 that the respondent completely agree with it (higher value).

The sample size establishes the error estimation of the sample. Since this is a critical aspect to be considered, it must be established a minimum sample size [90]. In this work the minimum sample size was defined as referred by Westland, in 2010 [91], and the value obtained was 200. Therefore, our sample of 252 validated answers did fulfill the imposed requirement.

## 4. Evaluation Models

In this section will be presented both measurement model (MM) and structural model (SM) used, as represented through Figure 2, which are composed by five constructs (Lean, Industry 4.0, Economic Sustainability, Environmental Sustainability, and Social Sustainability), each one measured by three indicators, totalizing 15 indicators, all measured in Likert scales (1–5). The factorial weights of each manifest variable (λ), of each coefficient estimation (β) and of each error (e) are obtained through the Maximum Likelihood Method [86–88].

It is mandatory to analyze the fit and validity of the identified variables, to assess the quality of the collected data, for further evaluation of the defined general model. Moreover, as a consequence perform the evaluation of the present work. The SEM method uses several different validation indexes, and in this study, it is considered the most common absolute, relative, parsimony-adjusted, population discrepancy and information theory-based indexes, whose specific reference is presented ahead of each corresponding index are summarized below.

- Absolute indices: these indices compare a specific model of adjustment with its saturated model.

  ○ $\chi^2/df$ ratio (chi-square and degrees of freedom ratio) [92,93];
  ○ Goodness of Fit Index (GFI) [92,93];

- Relative indices: relative adjustment indices compare the specific adjustment model to the worst possible adjustment (without relations between the manifested variables) and to the best possible adjustment (saturated model).

  ○　　Comparative Fit Index (CFI) [92,93];
  ○　　Tucker Lewis Index (TLI) [92];
  ○　　Incremental Fit Index (IFI) [93,94];

- Parsimony-adjusted indices: penalize the relative indices by complexity and perform an improvement in the model so as to bring it closer to the saturated model, through the inclusion of free parameters.

  ○　　Parsimony CFI (PCFI) [95];
  ○　　Parsimony GFI (PGFI) [95];

- Population discrepancy index: this index reflects the adjustment of the model at the sampling moments (means and sample variances) with the population moments (means and population variances) by the comparison effect.

  ○　　Root Mean Square Error of Approximation (RMSEA) [92];

- Information theory-based index: this index is pertinent to compare several alternative models with data adjustments.

  ○　　Akaike Information Criterion (AIC) [96].

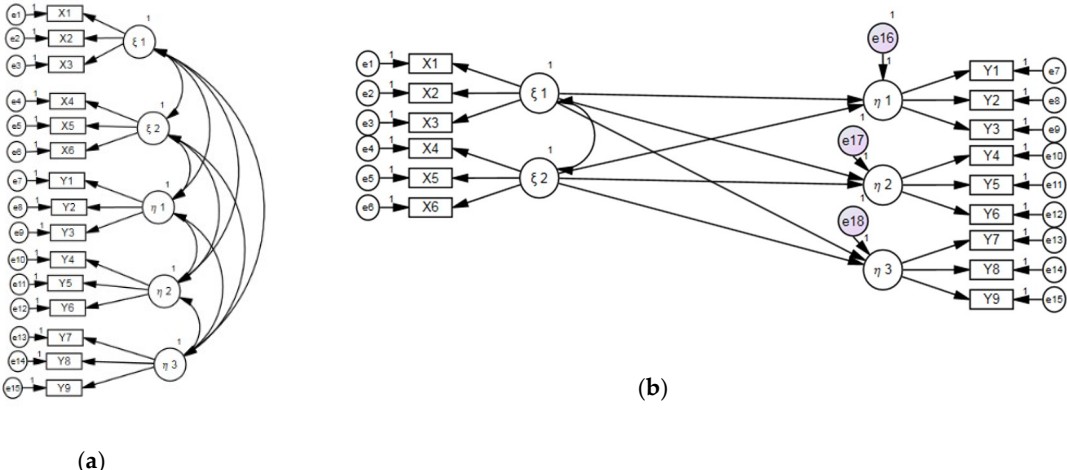

(a)

(b)

**Figure 2.** (**a**) Measurement model (**b**) structural model.

All the referenced indices are summarized in the Table 8, and the corresponding adjustment measures to achieve the indices values for a good fit. Additionally, it is presented the macros used in the software IBM SPSS Amos.

**Table 8.** Adjustment indices and measures.

| Adjustment Indices | Adjustment Measures | Macro in Amos SW | References |
|:---:|:---:|:---:|:---:|
| χ2/df | <3 | \cmindf | (Hu and Bentler, 1999), (Wei et al., 2010) [92,93]; |
| GFI | >0.9 | \gfi | (Hu and Bentler, 1999), (Wei et al., 2010) [92,93]; |
| CFI | >0.9 | \cfi | (Hu and Bentler, 1999) [92]; (Wei et al., 2010), (Singh, 2009) [93,97]; |
| TLI | >0.9 | \tli | (Hu and Bentler, 1999), (Singh, 2009) [92,97]; |
| IFI | >0.9 | \ifi | (Santora and Bentley, 1990), (Wei et al., 2010) [93,94]; |
| PCFI | >0.6 | \pcfi | (Mulaik et al., 1989) [95]; |
| PGFI | >0.6 | \pgfi | (Mulaik et al., 1989) [95]; |
| RMSEA | <0.08; $p > 0.05$ | \rmsea \pclose | (Hu and Bentler, 1999), (Wei et al., 2010) [92,93]; |
| AIC | Smaller than the independent model | \aic | (Schmitt, 2011) [96]. |

## 5. Results Discussion and Practical Implications

This section is divided into two subsections, the first one about the analysis of the mains results obtained, and the second one focused on the discussion on how these results may further support decision making in industrial companies and among their stakeholders. In the subsection about results discussion is first performed an estimation of the parameters of the models (MM and SM). Next, the parameters of the MM are analyzed and evaluated, and after an evaluation regarding the quality of the adjustment indices of the MM and the SM models is addressed.

### 5.1. Results Discussion

Regarding the estimates of the parameters of the models, using the SPSS Amos software, version 24 [89], with the Maximum Likelihood Method, the factorial and structural weights were obtained. In Figure 3a it is presented the relation between the constructs of MM, factorial weights, fit and errors. All constructs ($\xi 1$, $\xi 2$, $\eta 1$, $\eta 2$, $\eta 3$) are based on the manifest variables ($\xi 1$ = X1, X2, X3; $\xi 2$ = X4, X5, X6; $\eta 1$ = Y1, Y2, Y3; $\eta 2$ = Y4, Y5, Y6; $\eta 2$ = Y7, Y8, Y9) and, in this model, the constructs have a relation from all to all in order to provide the measurement evaluation in the MM. Figure 3b presents the SM, factorial weights, fit, and errors. For this model, the relations of the constructs are based on the initial general model (Figure 1), establishing the SM. Additionally, it was created a correlation between constructs LM ($\xi 1$) and I4.0 ($\xi 2$), because it seems an important issue, to be studied also.

Concerning the analysis and evaluation of the parameters of the MM, the results for LM ($\xi 1$) show good fit and validity of the indicators, due to the factorial weights being higher than 0.25 [92]. The most convergent indicators are the Failures (X3) with a factorial weight of 0.88, followed by product defects (X2) with a factorial weight of 0.74 and, finally, the pull production (X1) with a weight of 0.46. Good fit and validity have also been demonstrated in the indicators of the I4.0 construct ($\xi 2$), presented in decreasing order, Autonomous robots (X5), Big data (X4), and Digitalization (X6), with the factorial weight of 0.83, 0.82, and 0.56, respectively. Based on this data, it is possible to realize that the manifest variables that did show a higher influence for each of the defined exogenous constructs are: Failures (X3) for the LM construct ($\xi 1$), and Autonomous robots (X5) for the I4.0 construct ($\xi 2$).

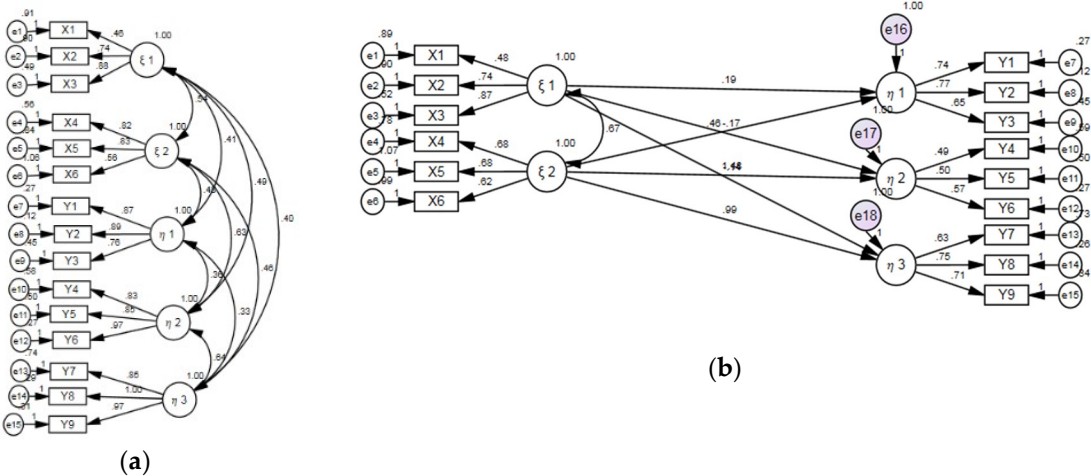

(a)

(b)

**Figure 3.** (**a**) Relation between the constructs of Measurement Model (MM), (**b**) Structural Model (SM), factorial weights, and fit.

The indicators Turnover (Y2), with a factorial weight of 0.89, Profits (Y1), with a factorial weight of 0.87, and Market share (Y3), with a factorial weight of 0.76, for Economic Sustainability ($\eta$1). In the Environmental Sustainability ($\eta$2) the higher indicator is Environmental practices with partners (Y6), with a factorial weight of 0.97, followed by Circular economy (Y5), with a factorial weight of 0.85, and Energy consumption (Y4), with a factorial weight of 0.83. Moreover, for Social Sustainability ($\eta$3), the indicators Work conditions (Y8), with a factorial weight of 1.00, Surrounding society (Y9), with a factorial weight of 0.97, and Salary remuneration (Y7), with a factorial weight of 0.85, also suggest a good fit and validity of the indicators of the endogenous constructs. Based on this data, it is possible to realize that the manifest variables that did show a higher influence for each of the defined endogenous constructs are: Turnover (Y2) for the Economic Sustainability ($\eta$1), Environmental practices with partners (Y6) for the Environmental Sustainability ($\eta$2), and Work conditions (Y8) for the Social Sustainability ($\eta$3).

The evaluation of the MM was performed, then the quality of the adjustment indices was obtained and the values are presented in Table 9.

**Table 9.** Adjustment validation of the MM. GFI: Goodness of Fit Index; CFI: Comparative Fit Index; TLI: Tucker Lewis Index; IFI: Incremental Fit Index; PCFI: Parsimony CFI; PGFI: Parsimony GFI; RMSEA: Root Mean Square Error of Approximation; AIC: Akaike Information Criterion.

| Adjustment Measures | Adjustment Obtained Value | Adjustment Criterion |
|---|---|---|
| $\chi^2/\mathrm{df}$ | 2.015 | <3 |
| GFI | 0.923 | >0.9 |
| CFI | 0.951 | >0.9 |
| TLI | 0.936 | >0.9 |
| IFI | 0.952 | >0.9 |
| PCFI | 0.724 | >0.6 |
| PGFI | 0.615 | >0.6 |
| RMSEA | 0.064 ($p = 0.058$) | <0.08; $p > 0.05$ |
| AIC | 241.163 < 1787.589 | Smaller than the independent model |

Due to the obtained value on the indexes, it is confirmed that all index values are within the defined criteria. After the evaluation of the MM, the SM was evaluated and, based on the defined quality of the adjustment indices, the values obtained are mentioned in Table 10.

**Table 10.** Adjustment validation of the SM.

| Adjustment Measures | Adjustment Obtained Value | Adjustment Criterion |
|---|---|---|
| $\chi^2/\mathrm{df}$ | 2.273 | <3 |
| GFI | 0.908 | >0.9 |
| CFI | 0.936 | >0.9 |
| TLI | 0.919 | >0.9 |
| IFI | 0.937 | >0.9 |
| PCFI | 0.740 | >0.6 |
| PGFI | 0.628 | >0.6 |
| RMSEA | 0.071 ($p = 0.006$) | <0.08; $p > 0.05$ |
| AIC | 262.657 < 1787.589 | Smaller than the independent model |

Based on the values shown in Table 9; Table 10 an analysis of the results of the validation of the theoretical models developed can be reached. When applying the Confirmatory Factor Analysis in the Measurement Model (MM), the values of the adjustment quality ($\chi^2/\mathrm{df}$ = 2.015, GFI = 0.923, CFI = 0.951, TLI = 0.936, IFI = 0,952, PCFI = 0.724, PGFI = 0.615, RMSEA = 0.064, and AIC = 241.163) confer a reliability and validity with a good fit. In the evaluation of the quality of the Structural Models (MEs), the results obtained ($\chi^2/\mathrm{df}$ = 2.273, GFI = 0.908, CFI = 0.936, TLI = 0.919, IFI = 0.937, PCFI = 0.740, PGFI = 0.628, and RMSEA = 0.071, AIC = 262.657), suggest that a good adjustment was reached.

Once again, all obtained index values are within the established range of the adjustment measures. Regarding the estimates for the parameters of the estimation of the structural relationships in the SM, the results shown in Table 11 were obtained.

**Table 11.** Estimates of the SM and synthetized frame of the hypothesis.

| Hypothesis | Exogenous Construct | Endogenous Construct | Est. | SE | CR | *p*-Value | Conclusion |
|---|---|---|---|---|---|---|---|
| H1. | Lean | Economic Sustainability | 0.187 | 0.133 | 1.405 | 0.16 | Not confirmed |
| H2. | Lean | Environmental Sustainability | −0.167 | 0.365 | −0.457 | 0.648 | Not confirmed |
| H3. | Lean | Social Sustainability | −0.142 | 0.280 | −0.508 | 0.611 | Not confirmed |
| H4. | Industry 4.0 | Economic Sustainability | 0.457 | 0.132 | 3.466 | <0.001 | Confirmed |
| H5. | Industry 4.0 | Environmental Sustainability | 1.482 | 0.477 | 3.108 | 0.002 | Confirmed |
| H6. | Industry 4.0 | Social Sustainability | 0.994 | 0.297 | 3.341 | <0.001 | Confirmed |

Concerning the analysis of the results of the assumptions of the SM, presented in Table 11, through the relationships established between the constructs are evaluated, and the hypotheses of the model are tested, as it is the main concern of this work. In the SM the hypotheses H4, H5, and H6 were confirmed, while the hypotheses H1, H2, and H3 were not confirmed (H1: $p = 0.16 > 0.05$, H2: $p = 0.648 > 0.05$, H3: $p = 0.611 > 0.05$). Additionally, a correlation (0.68) between Lean Manufacturing and Industry 4.0 is also confirmed in SM.

The results of our study, based on the developed model, and obtained through the Maximum Likelihood Method, show that: (1) it exists a strong relation between I4.0 and the three pillars of Sustainability, with a strongest factorial weight for Environmental Sustainability (1.482), followed by Social Sustainability (0.994), and the lowest for Economic Sustainability (0.457); (2) it is not confirmed that exists a relation between LM and Sustainability; (3) cumulatively, it was found out that exists a correlation between LM and I4.0.

*5.2. Practical Implications*

The results of this study are not totally aligned with the initial expectations regarding the relation between LM and Sustainability stressed by several authors, for instance the authors reviewed in Section 2.1. Although, it does not mean that no relation at all does exist. However, this study does not confirm that relation. This is possibly explained due to the fact that LM focuses its attention on a current state of a company, without a concern regarding a global, integrative and transformative vision of companies, and subject to dynamic and turbulent environments, which is a general characteristic of them, namely industrial ones. This is due to the fact that LM is based on a "linear", non-holistic thinking, thus not taking into account the need to include other more sustainable paradigms. In terms of practical implications for the companies, now they have a structured study to be considered when faced with a decision making process to implement LM. From this moment, the fact is that the companies know that it is not confirmed that LM is related to Sustainability.

The other part of the results, regarding the relation between I4.0 and Sustainability, was confirmed to be aligned with most of the researchers' contributions in topics of this field, for instance, the authors reviewed in Section 2.2. However, it does not mean there do not exist further concerns about I4.0 implications in the three pillars of Sustainability, namely in the social one, principally for the employees. In terms of practical implications for the companies and their stakeholders, this study assures the existence of relation between I4.0 and Sustainability. This can mean that companies have now a stronger knowledge to further decide about I4.0 implementation and its implications in sustainability.

## 6. Conclusions

In this paper, after an introduction to the principal goals of the work and the explanation of the meaning of the three main elements underlying this work (LM, I4.0, and Sustainability), was done a literature review mainly focused in the influence of LM in Sustainability and of I4.0 in Sustainability.

Based on that revision, and as far as our knowledge, were obtained the following three main remarks: (1) none of the researchers in their works did cover all the considered main influence criteria exposed; (2) none of the works analyzed treats this subject through SEM; and (3) some criteria' influence are more considered than others, for instance, for few of them were not found any reference. Then, it was validated that this work focuses a different approach than the literature analyzed, and it was possible to put forward the main constructs and the manifested variables used in our proposed model. To address that gap, it was proposed a structural equation model, based on two exogenous constructs (LM and I4.0) and three endogenous constructs (EcS, EnS, and SoS), each construct composed by three manifest variables, and with six hypotheses, to quantitatively measure the effects of LM and I4.0, in Sustainability. Moreover, in order to statistically validate such hypotheses, a set of 252 valid questionnaires from industrial companies of Iberian Peninsula (Portugal and Spain) were analyzed.

The validation of the MM was obtained through the application of the Confirmatory Factor Analysis and the corresponding values of the adjustment quality conferred a reliability and validity with a good fit. Concerning the assumptions of the SM values analysis, the relationships established between the constructs were evaluated and the hypotheses of the model tested, as it was a main concern of this work. Then, the hypotheses H4, H5, and H6 were confirmed, while the hypotheses H1, H2, and H3 were not. Additionally, a correlation between LM and I4.0 was also confirmed.

As a global conclusion, the results obtained through the study carried out enable to state that exists a relation between I4.0 and Sustainability, and a not confirmed relation between LM and Sustainability. These conclusions can contribute as an important decision support for the industrial companies and its stakeholders, even because not all the results are in line with other opinions and studies. Moreover, this can mean that companies have now a stronger knowledge to further decide about the implementation of LM and I4.0, and its implications in Sustainability.

The results of this study are limited by the region object, in this case the Iberian Peninsula, and for industrial companies. For that reason, further studies should consider other countries. Also, because social sustainability is an important concern for the future of society, namely, concerning the influence

of I4.0, more variables will be considered in the presented model to evaluate more deeply its influence in the ambit of social sustainability.

**Author Contributions:** All authors contributed equally to this work and approved the final manuscript.

**Funding:** This research was funded by FCT—Fundação para a Ciência e Tecnologia, grants numbers UID/CEC/00319/2019, and UID/615/2019.

**Acknowledgments:** This work has been supported by FCT—Fundação para a Ciência e Tecnologia within the Projects Scope: UID/CEC/00319/2019, and UID/615/2019. Moreover, the authors thank the 2100 Projects Association for the monetary support to develop this study, the three experts that contributed for survey validation, and also the enriching improvement suggestions of the Journal reviewers.

**Conflicts of Interest:** The authors declare no conflict of interest.

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
