# Peer review of "Evaluation of the Relation between Lean Manufacturing, Industry 4.0, and Sustainability"

_sustainability, doi:10.3390/su11051439_

Round 1
Reviewer 1 Report
This paper presents a survey study to correlate Lean Manufacturing and Industry 4.0 for Sustainability from three pillars, economic, environmental and social.
- Although the paper presents an extensive review of related works, most of the citations are not described.
- The main contribution of the paper is the correlation between Lean Manufacturing and Industry 4.0 and two methods are presented. However, the methodology should be more detailed. For instance, why you have chosen the adjustment measures shown in table 9, how you worked with the validation indexes with the software and why you decided to use those exactly.
- Why LM does not contribute to Sustainability? Isn't it avoid wasteful resources? How you can explain this fact?
- I guess from the software you can extract additional results. Please, extend the results section.
Author Response
Dear Reviewer,
Please find attached our answers to your fruitful comments, along with the revised manuscript.
Kind regards,
The authors

Reviewer 2 Report
Dear Authors,
thank you for the possibility to read the paper. Please find below several remarks for improvement of the quality of the paper:
Abstract. It is mentioned that results contribute to decision making of industrial companies and their stakeholders. Could you please shortly mention: In which way could they support decision making?
Introduction. Comprehensive and good quality section. All relevant elements that should be included in the Introduction are found.
Literature Review. Very concentrated section. The Authors could allow for themselves to be more descriptive and provide more explanations and observations in the text. Moreover, with the purpose to show the contribution of the paper to world science and novelty of the work, i would propose to expand Tables 1 to 6 with additional columns, such as Time period covered by studies, Methodology applied, Economic sector, Country, etc. In addition, i would propose to explain column Influence of Tables 4 to 6 in a way it is done in tables 1 to 3. For example, increase in profit, etc.
Research Model. Clearly written section. As a Reader i could not understand what is exogenous and endogenous constructs in your research? maybe it is possible to write several sentences on this.
Evaluation Models. As a Reader i missed explanations of abbreviations that are in Figure 2 (e1, X1, etc..).
Results and Discussion. Very technical presentation of results, therefore, my recommendation is to explain results in the text much more comprehensive, i.e. what is the meaning of numbers presented in Figure 3? Short analysis of Figure 3 should be given in the text.
Implications. I would propose to include additional section of Implications. Here, the Authors could provide discussion on how (the way) their results support decisions of the industrial companies and their stakeholders? What industrial companies could learn, how they should shape their policy, activities, etc. This section could be useful for understanding how results could be adapted to practice.
Conclusions. I would think that this is the weakest section of the paper. Results should be supported by numbers calculated. What conclusions could be made from your research? For example, The analysis of SM model analysis showed that...; The review of literature allowed summarizing the key KPIs, which are...etc.
References. Very long list and this is an advantage, because shows a strong background of the research. However, this strong background was disclosed in very concentrated manner in Literature Review.
Best regards,
Reviewer
Author Response

(The authors gave the same response as above.)

Round 2
Reviewer 1 Report
Most results are within the conclusion section instead of inside the results one. The conclusion should give just a brief description of the results.
You say "none of the researchers" and other statements from line 404. Those are strong statements. You should say as far as your knowledge or a similar phrase. Although you have explored the literature, you cannot assume you have checked it all.
Author Response
Dear Reviewer,
The authors’ thank you very much for your comments and suggestions provided for the improvement of the paper, and please find attached the answers letter.
Best regards,
The authors

Reviewer 2 Report
Dear Authors,
i would recommend this paper to be published in the journal.
BR
Reviewer
Author Response
Dear Reviewer,
The authors’ thank you very much for your comment and decision provided, and please find attached the answers letter.
Best regards,
The authors
